# Identification, Genetic Characterization, and Pathogenicity of Three Feline Herpesvirus Type 1 Isolates from Domestic Cats in China

**DOI:** 10.3390/vetsci11070285

**Published:** 2024-06-25

**Authors:** Mingliang Deng, Haiyang Liang, Yue Xu, Qiwen Shi, Fang Bao, Caiying Mei, Zhihong Dai, Xianhui Huang

**Affiliations:** 1Guangdong Provincial Key Laboratory of Veterinary Pharmaceutics Development and Safety Evaluation, College of Veterinary Medicine, South China Agricultural University, Guangzhou 510642, China; dengmingliang@bio-ss.net; 2Ningbo Sansheng Biological Technology Co., Ltd., Ningbo 315000, China

**Keywords:** isolated feline herpesvirus, clinical samples, phylogenetic analysis, histopathological changes, pathogenicity

## Abstract

**Simple Summary:**

Feline herpesvirus (FHV-1) is a common virus in cats that causes respiratory and ocular diseases. This research aimed to study FHV-1 by isolating it from nasal and ocular samples collected from symptomatic cats and evaluating its characteristics and virulence. We collected 35 samples from symptomatic cats with respiratory infections where FHV-1 was detected by polymerase chain reaction (PCR). The virus was isolated using feline kidney (F81) cell lines, identified using standard laboratory techniques, and analyzed for genetic characteristics. We found three different strains of FHV-1 that shared high homology among themselves and with domestic isolates and FHV-1 viruses isolated worldwide. However, these strains caused varying disease severities in their host. One strain (FHV-A1) was highly virulent, resulting in high morbidity and mortality. Our work enhances our understanding of the genetic diversity and pathogenesis of Chinese FHV-1 strains and also underscored FHV-A1 isolate as a potentially ideal candidate for establishing a challenge model and as a potential vaccine strain, which lays the groundwork for vaccine development.

**Abstract:**

(1) Background: Feline herpesvirus (FHV-1) is a significant pathogen in cats, causing respiratory and ocular diseases with consequential economic and welfare implications. (2) Methods: This study aimed to isolate and characterize FHV-1 from clinical samples and assess its pathogenicity. We collected 35 nasal and ocular swabs from cats showing symptoms of upper respiratory tract infection and FHV positivity detected by polymerase chain reaction (PCR). Viral isolation was carried out using feline kidney (F81) cell lines. Confirmation of FHV-1 presence was achieved through PCR detection, sequencing, electron microscopy, and indirect immunofluorescence assay. The isolated strains were further characterized by evaluating their titers, growth kinetics, and genetic characteristics. Additionally, we assessed the pathogenicity of the isolated strains in a feline model, monitoring clinical signs, viral shedding, and histopathological changes. (3) Results: Three strains of FHV-1 were isolated, purified, and identified. The isolated FHV-1 strains exhibited high homology among themselves and with domestic isolates and FHV-1 viruses from around the world. However, they showed varying degrees of virulence, with one strain (FHV-A1) causing severe clinical signs and histopathological lesions. (4) Conclusion: This study advances our understanding of the genetic and pathogenic characteristics of FHV-1 in China. These findings underscore FHV-A1 isolate as a potentially ideal candidate for establishing a challenge model and as a potential vaccine strain for vaccine development.

## 1. Introduction

Feline herpesvirus type 1 (FHV-1), a member of the *Alphaherpesvirinae* subfamily within the Herpesviridae family [1], constitutes a significant viral pathogen responsible for acute and chronic upper respiratory tract diseases, conjunctivitis, and keratitis in domestic cats [2]. Rapidly disseminating through direct contact with infected secretions or contaminated objects, this virus is highly contagious within feline populations. Despite advancements in vaccines and antiviral therapies [3], FHV-1 continues to inflict substantial morbidity in cats globally [4], imposing significant economic burdens on the pet industry and posing challenges in animal shelters and breeding facilities [5,6].

The clinical manifestation of FHV-1 infection can vary from mild, self-limiting cases to severe, life-threatening conditions, contingent upon factors such as the virulence of the viral strain, the immune status of the host, age, and environmental conditions [7]. Common clinical presentations include sneezing, fever, nasal and ocular discharge, conjunctivitis, keratitis, lethargy, anorexia, and coughing [8]. FHV-1 exhibits the capability to establish latency within the trigeminal ganglia [9], with periodic reactivation leading to recurrent episodes of clinical disease, particularly in instances of stress or immunosuppression. This ability to establish lifelong latent infections complicates disease management and control efforts [10].

FHV-1 harbors a double-stranded DNA genome spanning approximately 135 kilobases, exhibiting relative conservation among isolates, yet it is susceptible to genetic variations that could impact pathogenicity and antigenicity [5,6]. Recent investigations underscore the genetic diversity within FHV-1 strains, proposing that variations within specific viral genes might dictate pathogenicity, tissue tropism, and antigenicity [11]. Notably, *glycoproteins B*, *D*, and *E* (*gB*, *gD*, and *gE*) and the thymidine kinase (*TK*) gene have garnered attention due to their pivotal roles in viral entry, replication, and evasion of the host immune response [12]. However, the association between genetic variations within these regions and the clinical manifestations of FHV-1 infection remains elusive [10]. In addition, the genetic and pathogenic characterization of FHV isolates from domestic cats in China remains scarce, posing a limitation in the development of more efficacious vaccines and therapeutic interventions for managing FHV-1 infections among felines in China. Moreover, the low seroprevalences of FHV and FCV in vaccinated cats indicated that the abroad commercial triple-inactivated vaccine (Fel-O-Vax^®^ PCT vaccine, Zoetis, Parsipanny, NJ, USA) utilized in China could not resist the infection of domestic wild strains of FHV and FCV, which highlights the urgency for more effective vaccines [13].

To address these challenges, this study aimed to isolate and characterize FHV strains isolated from nasal and ocular swabs of cats in China, labeled as FHV-A1, FHV-C8, and FHV-D3. Molecular techniques, including polymerase chain reaction (PCR), sequencing, and phylogenetic analysis, were primarily employed to explore the genetic characteristics of these three strains. Furthermore, we evaluated the pathogenicity of these strains in a feline model to glean insights into their virulence and potential impact on disease outcomes. The findings of this study enrich our understanding of the diversity and pathogenesis of Chinese FHV-1 strains and hold implications for the enhancement of vaccines and therapeutic strategies geared towards managing FHV-1 infections in cats in China. Finally, FHV-A1 isolate underscores its potential as an ideal candidate for establishing a challenge model and as a potential vaccine strain, which lays the groundwork for vaccine development.

## 2. Materials and Methods

### 2.1. Ethics Statement

This study was conducted in accordance with the recommendations outlined in the Regulations for the Administration of Affairs Concerning Experimental Animals by the Ministry of Science and Technology of China. The study protocol was approved by the Scientific Ethics Committee of Ningbo Sansheng Biological Technology Co., Ltd. (permit number: NBSS-AEC-202005001).

### 2.2. Clinical Samples and Cells

Thirty-five nasal and ocular swabs were collected from symptomatic cats preliminarily diagnosed with FHV infection using real-time PCR testing in pet hospitals in Beijing City and various provinces (Guangdong, Heilongjiang, Jiangsu, Sichuan, Zhejiang) in China from 2019 to 2020 (see Appendix A). Feline kidney (F81) cells (ATCC CL-0081) were cultured and maintained by Ningbo Sansheng Biological Technology Co., Ltd (Ningbo, China). These F81 cells were propagated in Dulbecco’s modified Eagle’s medium (DMEM) (Gibco, Grand Island, NY, USA) supplemented with 5% fetal bovine serum (FBS) (Hyclone, Logan, UT, USA), 100 μg/mL streptomycin, and 100 IU/mL penicillin and incubated at 37 °C with 5% CO_2_.

### 2.3. Virus Isolation

Approximately 2.0 mL of DMEM (containing 100 U/mL penicillin and 100 μg/mL streptomycin) was added to the nasal and ocular swabs. The samples were vortexed and then centrifuged at 8000 rpm for 10 min. The supernatant was collected and filtered through a 0.22 µm filter. The filtrate was used for PCR detection using specific primers (see Appendix A) for FPV, FCV, and FHV, as well as for virus isolation.

Nasal and ocular swabs that tested positive for FHV by PCR were inoculated onto F81 cells. The presence or absence of cytopathic effects (CPE) was evaluated daily. When CPE were observed, the cell culture was harvested once the CPE reached over 80%. The harvested culture was subjected to two freeze–thaw cycles at −70 °C, followed by centrifugation at 4000 rpm for 5 min. The supernatant was then subjected to virus purification and identification. If no CPE appeared after 4 days of inoculation, the cell plates underwent two freeze–thaw cycles. Subsequently, the cell culture was centrifuged at 4000 rpm for 5 min, and the supernatant was subjected to further passaging.

### 2.4. Virus Plaque Purification

The supernatant showing CPE was subjected to virus plaque purification by serially diluting the supernatant tenfold to 10^−7^. Dilutions of 10^−5^, 10^−6^, and 10^−7^ were inoculated onto F81 cell monolayers in 6-well plates. After incubation at 37 °C for 1 h, the medium was removed, and the cells were overlaid with DMEM containing 2% low melting point agarose (melted at 60 °C in a water bath) and 2% FBS. The plates were then incubated at 37 °C with 5% CO_2_ for 2 days to observe plaque formation.

Five individual plaques/clones were picked for each virus under a microscope using a pipette tip and resuspended in DMEM medium in the first round of plaque purification. The virus clones were subsequently inoculated onto F81 cells, and the titers of these clones were determined. The clone with the highest virus titer was chosen for further plaque purification, and this process was repeated for two additional rounds. After three rounds of plaque purification, a FHV isolate was successfully obtained.

### 2.5. Molecular Detection and Indirect Immunofluorescence Assay (IFA)

Viral DNA and RNA were extracted from purified virus by using a FastPure Viral DNA/RNA Mini Kit (Vazyme, Catalog no. RC311-01, Nanjing, China) according to the manufacturer’s instructions and subjected to PCR analysis using specific primers designed for detection of FHV, FCV, and FPV. The primer sequences and PCR conditions are shown in Appendix A.

The F81 cells were cultured in 6-well plates until a monolayer was formed, and then the virus was inoculated onto the cells, with two replicates for each sample, at a multiplicity of infection (MOI) of 0.01. Following 2 days of incubation at 37 °C with 5% CO_2_, IFA was performed following standard protocols. The cell monolayers were fixed with 4% paraformaldehyde solution for 30 min and permeabilizated with 0.1% Triton X-100 for 15 min. Primary antibody (mouse anti Feline Herpes Virus Type 1 antibody clone FHV7-7C, Bio-Rad, Catalog no. MCA2490, Hercules, CA, USA) specific to *gB*, *gC*, *gI*, and *gD* of FHV-1 was added at 1:2000 dilution in 1% BSA for 1 h and washed with PBS three times. Finally, secondary antibody (iFluor™ 488 Conjugated Goat Anti-Mouse IgG Antibody, Hangzhou Huabio, Catalog no. HA1125, Hangzhou, China) was added at a 1:1000 dilution in 1% BSA for 1 h. After removal of the secondary antibody solution, the cells were washed three times with PBS and examined under a fluorescence microscope.

### 2.6. Electron Microscopy

After propagating the purified virus, the culture supernatant underwent centrifugation at 8000 rpm for 5 min to remove cell debris. Subsequently, the supernatant was ultracentrifuged over a 30% sucrose cushion at 25,000 rpm for 1 h. The virus particles were then sent to the Center for Excellence in Brain Science and Intelligence Technology for examination by electron microscopy.

### 2.7. Virus Titer and Virus Growth Curve

The virus was serially diluted tenfold in DMEM, ranging from 10^−3^ to 10^−9^, and it was inoculated onto F81 cell monolayers in 96-well plates. Each dilution was tested in 6 wells, with 0.1 mL of inoculum per well. The plates were incubated at 37 °C with 5% CO_2_ for 4 days, during which we observed and recorded the CPE. We determined the virus titer using median tissue culture infectious dose (TCID_50_) according to the Reed–Muench method [14].

We inoculated the virus onto F81 cell monolayers in 6-well plates at a multiplicity of infection (MOI) of 0.01. After adsorption at 4 °C for 1 h, we removed the supernatant and washed the cells twice with PBS. We then added 2 mL of culture medium to each well and incubated the plates at 37 °C with 5% CO_2_. We collected the supernatants and cells at 12, 24, 36, 48, 60, and 72 h post-infection. Each time point had three replicate wells. The samples underwent two freeze–thaw cycles and were centrifuged at 8000 rpm for 5 min. We titrated the supernatants using TCID_50_ and plotted a virus growth curve [15].

### 2.8. Sequencing and Phylogenetic Analysis

Viral DNA was extracted from cell cultures exhibiting CPE by using FastPure Viral DNA/RNA Mini Kit (Vazyme, Catalog no. RC311-01) according to the manufacturer’s instructions and sent to Beijing Novogene Bioinformatics Technology Co., Ltd. (Beijing, China), for whole-genome sequencing using next-generation sequencing technology.

The obtained sequences were aligned using the BLAST tool on the NCBI website. Multiple sequence alignment with published sequences of FHV reference and vaccine strains (see Appendix A) was conducted with Clustal W algorithm in MEGA7.0 software. Sequence homologies were analyzed using BioEdit software 7.2 [16]. Phylogenetic trees of the nucleotide sequences for the *gB*, *gD*, *gE*, and *TK* genes were constructed using the Neighbor-Joining method in MEGA7.0 software, with a bootstrap value of 1000 [17].

### 2.9. Pathogenicity Evaluation of FHV Isolates in Kittens

Twenty healthy 8–12-week-old kittens free of antigens and antibodies to FHV-1, feline calicivirus (FCV), and feline panleukopenia virus (FPV) were randomly divided into four groups, with five kittens in each group.

The kittens in Groups 1–3 were inoculated intranasally with 2 mL (10^7.0^ TCID50/mL) of the FHV-A1, FHV-C8, and FHV-D3 isolates, respectively. Group 4 served as the negative control, inoculated with 2 mL of DMEM.

We measured and recorded the baseline rectal temperature of the kittens daily for three days before inoculation. After inoculation, we continued to measure rectal temperature daily for 14 consecutive days. We observed and scored clinical signs, such as depression, conjunctivitis, blepharospasm, ocular discharge, nasal discharge, sneezing, nasal congestion, and coughing daily, following the references [18,19], starting on the day of inoculation and continuing for 14 days. Appendix A details the scoring criteria for these clinical symptoms. We performed immediate necropsies on any kittens that died during the experiment. At 14 days post-inoculation, we euthanized and necropsied the surviving kittens to examine gross pathological changes. We fixed the lung tissue of the kittens in buffered 4% formalin, embedded it in paraffin wax, sectioned it, and stained it with hematoxylin and eosin for histopathological examination.

We collected ocular and nasal mixed swabs daily for 14 days post-inoculation to detect the excreted virus through virus isolation [18].

### 2.10. Statistical Analysis

We analyzed and graphically presented the experimental data using GraphPad Prism 8 software. We expressed numerical values as mean ± standard deviation (mean ± SD).

## 3. Results

### 3.1. Isolation and Purification of the FHV

We processed nasal and ocular swab samples from symptomatic cats and subjected them to PCR detection for FPV, FCV, and FHV. The results indicated that the single-antigen positivity rates were 71.4% (25/35) for FHV, 34.3% (12/35) for FCV, and 0% (0/35) for FPV. The double-antigen positivity rate for FHV and FCV was 14.3% (5/35). We inoculated F81 cells with five FHV-positive samples that had Ct values less than 28 in the real-time PCR test performed at the pet hospital. Three of these samples exhibited CPE in the first passage between 36 and 60 h (Figure 1), with nearly complete cell detachment after 60 h.

We harvested the three FHV isolates and subjected them to virus plaque purification. After three rounds of purification, we obtained three purified FHV isolates, designated as FHV-A1, FHV-C8, and FHV-D3.

### 3.2. Identification and Characterization of the FHV

We extracted DNA and RNA from the purified viruses to identify FHV, FCV, and FPV using PCR. Our results demonstrated that all three isolates yielded expected amplicons with a length of 1269 bp (*gD* gene) when amplified by FHV primers, while no amplicons were detected for other viruses (Figure 2; for the original gel image, refer to Appendix A). This confirms that all three isolates were FHV.

We conducted an IFA assay on F81 cells inoculated with FHV isolates using a Feline Herpes Virus Type 1 monoclonal antibody as the primary antibody. Our findings revealed distinct specific green fluorescence in the cytoplasm of infected cells from all three isolates and the positive control, while no specific fluorescence was observed in the normal cell control (Figure 3). This further supports the identification of all three isolates as FHV.

We propagated and ultrapurified the purified viruses and examined them under electron microscopy. Our results revealed round virus particles with a diameter ranging from approximately 150 to 160 nm (Figure 4), consistent with the expected size (150~200 nm) and morphology of FHV particles.

The titers of FHV-A1, FHV-C8, and FHV-D3 were determined to be 10^7.57^ TCID_50_/mL, 10^7.17^ TCID_50_/mL, and 10^7.13^ TCID_50_/mL, respectively, with FHV-A1 exhibiting the highest titer. We evaluated the viral growth kinetics of the FHV-A1, FHV-C8, and FHV-D3 strains in F81 cells, and the multiple-step growth curve indicated no significant difference in the growth kinetics between FHV-C8 and FHV-D3. However, FHV-A1 replicated faster and reached a higher titer during the plateau phase. The highest titer for FHV-A1 was 10^7.62^ TCID_50_/mL at 60 h post-infection, which was significantly higher than FHV-C8 and FHV-D3, with titers of 10^7.17^ TCID_50_/mL and 10^7.08^ TCID_50_/mL, respectively (Figure 5).

### 3.3. Sequencing and Phylogenetic Analysis

We aligned the whole-genome sequences of FHV obtained through next-generation sequencing using the Blast tool, confirming that all three isolates were indeed FHV. Additionally, we conducted sequence alignment of the *gB*, *gD*, *gE*, and *TK* gene sequences of the three FHV isolates with domestic isolates and FHV-1 viruses isolated worldwide. Our analysis revealed that the three isolates shared 100% identity for *gB*, *gD*, and *TK* gene nucleotide and amino acid sequences among the strains. Furthermore, they demonstrated high homology (above 99.4%) of *gB*, *gD*, *gE*, and *TK* gene sequences with both domestic isolates and FHV-1 viruses isolated worldwide (Table 1).

Phylogenetic analysis of the nucleotide sequences of the *gD* and *gE* genes revealed that the three FHV isolates clustered together in the same branch alongside both domestic isolates of FHV-1, FHV-1 viruses isolated globally, and vaccinal strains (Figure 6), indicating a close relationship with these strains. However, when considering phylogenetic trees based on the *gD* and *gE* genes, the three isolates demonstrated a lower relationship with canine herpesvirus type 1 (CHV-1) isolates. The phylogenetic analysis of the nucleotide sequences of the *gB* and *TK* genes (refer to Appendix A) aligned consistently with the analysis based on the *gD* and *gE* genes.

### 3.4. Pathogenicity Evaluation of FHV Isolates in Kittens

#### 3.4.1. Clinical Sign Observation, Gross Pathology Examination, and Histopathological Changes

To assess the virulence of the FHV isolates, we randomly divided FHV-1, FCV, and FPV-free kittens into four groups and inoculated them with FHV-A1 (Group 1), FHV-C8 (Group 2), FHV-D3 (Group 3), and DMEM (Group 4). During the 14 day observation period post-inoculation, no obvious clinical signs were observed in the DMEM group, and their rectal temperatures remained within the normal ranges (Figure 7).

All five kittens in Group 1 exhibited fever (exceeding 1 °C compared to baseline) on multiple occasions: four, two, two, four, and two times. In Group 2, three kittens experienced fever on three, two, and three occasions, while two kittens in Group 3 exhibited fever on at least two occasions.

Within Group 1, one kitten displayed mild conjunctivitis, minor nasal discharge, mild nasal congestion, and sneezing on day 3 post-infection. By day 4 post-infection, three kittens exhibited observable nasal discharge with mild nasal congestion, and one kitten showed mild conjunctivitis and ocular discharge. All kittens in Group 1 displayed varying degrees of ocular and upper respiratory tract clinical signs by day 5 post-infection. This included mucopurulent ocular and nasal discharges (Figure 8), along with typical upper respiratory tract symptoms such as sneezing, coughing, conjunctivitis, and blepharospasm starting from day 6 post-infection. Three kittens succumbed to the infection on days 8, 9, and 12 post-infection, individually (Figure 9). Detailed scores for ocular and upper respiratory tract clinical signs for each animal are provided in Appendix A, respectively.

In Group 2, one kitten exhibited minor nasal discharge with mild nasal congestion on day 2 post-infection. By day 4 post-infection, it displayed mild conjunctivitis, blepharospasm, and ocular discharge, along with moderate nasal discharge and congestion. Three kittens within this group demonstrated varying degrees of ocular and upper respiratory tract signs from days 5 to 7 post-infection, peaking in severity on days 7 to 10 post-infection. These signs included mucopurulent ocular and nasal discharges, sneezing, coughing, conjunctivitis, and blepharospasm. Two kittens succumbed to the infection on days 8 and 11 post-infection, individually (Figure 9).

In Group 3, the kittens exhibited milder clinical signs compared to Group 1 and 2. Only two kittens displayed severe clinical signs of FHV infection, with one kitten passing away on day 12 post-infection (Figure 9).

Table 2 provides a summary of the typical clinical signs of FHV infection observed in the kittens infected with the FHV isolates, while Figure 9 illustrates the dynamics of mortality.

To further explore the pathogenicity of the three FHV isolates, we conducted an examination of pathological lesions. We observed significant pathological changes during the necropsy of kittens exhibiting severe clinical signs of FHV infection. These changes included pulmonary hemorrhage, the presence of a substantial amount of yellow mucus in the trachea, and hemorrhage and enlargement of the mesenteric lymph nodes, along with brain hemorrhage (Figure 10A–D). In contrast, kittens in the negative control group showed no pathological changes (Figure 10E–H). The histopathological examination revealed severe or mild interstitial pneumonia in the lungs of kittens displaying severe clinical signs of FHV infection, whereas no pathological lesions were evident in the lungs of kittens in the negative control group (Figure 11).

#### 3.4.2. Virus Shedding

We monitored viral shedding in ocular and nasal swabs daily following inoculation (Table 3). Our findings revealed that 100% (5/5) of kittens in the FHV-A1, FHV-C8, and FHV-D3 challenge groups shed the virus on days 3, 4, and 5 post-infection, respectively. The FHV-A1 and FHV-D3 groups continued shedding the virus until the end of the experiment, while the FHV-C8 group shed the virus up to day 12 after inoculation. Conversely, no virus was detected in kittens inoculated with DMEM. Appendix A provides the results of virus isolation from nasal and ocular swabs for each individual kitten.

## 4. Discussion

Feline herpesvirus (FHV-1), also known as feline rhinotracheitis virus, is a highly contagious viral pathogen affecting domestic cats worldwide. It belongs to the family Herpesviridae and is a leading cause of respiratory and ocular diseases in cats. Understanding its epidemiology, vaccination strategies, and treatment options for FHV-1 is paramount for effective management and containment of its dissemination [5]. Hence, a thorough analysis of pathogenicity and genetic variations of current variants becomes essential for devising control strategies. In this investigation, we isolated and identified FHV strains within China.

Our study tested nasal and ocular swabs for FPV, FCV, and FHV via PCR detection. The result showed that the single-antigen positivity rate for FHV was 71.4%, which is higher than that of a previous report on molecular investigation of cat viral infectious diseases in China from 2016 to 2019 (16.3%) [13] and a previous study of molecular prevalences in cats in Southern Italy (9.05%) [20]. A possible reason is that the nasal and ocular swabs used in our study were preliminarily collected from symptomatic cats and have been diagnosed with FHV infection using real-time PCR testing in pet hospitals. Previous studies utilized various samples collected from clinically diseased cats or dead cats with unknown pathogens [13,20]. FPV was not detected in our study as nasal and ocular swabs were used and since FPV is a parvovirus that causes enteritis and panleukopenia in domestic and wild cats, who mainly shed the virus in feces [21]. In addition, the double-antigen positivity rate for FHV and FCV was 14.3% as FHV and FCV are the main pathogens of upper respiratory tract infection in cats [22] and also because they are commonly co-infected as seen in previous reports [13,20].

The multiple-step growth kinetics of the FHV-A1, FHV-C8, and FHV-D3 strains in F81 cells indicated no significant difference in the growth kinetics between FHV-C8 and FHV-D3. FHV-A1 replicated faster and reached a higher titer during the plateau phase. The peak titer for FHV-A1 was 10^7.62^ TCID_50_/mL at 60 h post-infection. The growth kinetics and peak titer of the strains were consistent with previous reports that performed on CRFK cells [15,23], which indicated a similar adaptation of these two cell lines for FHV.

A systematic evolutionary scrutiny of the whole FHV genome revealed a remarkable degree of conservation, with only 0.01% diversity noted among different isolates [24]. A prior global study on FHV-1 isolates disclosed an overall genomic interstrain genetic distance of 0.093% [12]. Recent reports also confirmed high sequence homology among domestic and global FHV isolates [17]. Sequence analysis of the three isolated FHV strains in our study unveiled a 100% homology of *gB*, *gD*, and *TK* genes among the isolates. Furthermore, these three isolated FHV strains exhibited a high identity (99.4~100%) with isolates of FHV-1 from domestic cats and globally isolated FHV-1 viruses, consistent with earlier findings [17].

We conducted phylogenetic analyses of FHV utilizing various genes, including *gB*, *gD*, *gE*, *TK*, or whole genome sequences [8,25]. Previous reports have indicated a close relationship among FHV-1 strains and a low relationship with CHV-1 isolates. The phylogenetic analysis conducted in our present study, incorporating domestic and global strains, aligns with these prior findings, suggesting the stability of FHV immunogenicity and probable host selectivity. Notably, the *gB* and *gD* proteins serve as major immunogenic proteins in herpesviruses, crucial for inducing neutralizing antibodies. The high conservation of these genes likely contributes significantly to the virus’s immunogenic stability. However, the commercial vaccine (Fel-O-Vac vaccine, Zoetis) utilized in China could not resist the infection of the domestic wild strains of FHV and FCV as demonstrated by the low seroprevalences of FHV and FCV in vaccinated cats [13]. This indicates that other proteins, besides *gB* and *gD*, may contribute to immunogenecity. Additionally, *gB*, *gD*, *gE*, and the *TK* gene play pivotal roles in viral entry, replication, and evasion of the host immune response. This underscores the low homology and relationship between FHV-1 and CHV-1 isolates, potentially leading to differing host selectivity [12].

Following challenge infection with FHV isolates, upper respiratory tract and ocular diseases are commonly observed [26]. The severity of respiratory and ocular disease in cats infected with FHV-1 can vary considerably. Some animals exhibit mild ocular discharge, blepharospasm, and occasional sneezing, while others display severe ocular lesions [15]. Our pathogenicity assessment revealed that the three FHV isolates exhibited varying degrees of virulence, eliciting typical clinical signs of FHV infection in affected cats, including mucopurulent ocular and nasal discharges, sneezing, coughing, conjunctivitis, and blepharospasm. Additionally, our study unveiled that different viral isolates can induce varying disease severities in hosts, despite minimal genomic variation being observed among these isolates based on sequence and phylogenetic analysis. This finding aligns with previous reports, suggesting that the spectrum of host disease severity associated with FHV-1 is likely not primarily linked to viral genomic variations but rather to host response and/or other factors [11].

Upon pathogenicity evaluation, the FHV-A1 strain demonstrated the highest virulence, with 100% of kittens exhibiting severe respiratory and ocular disease, and 60% of animals succumbing during the experiment. The morbidity and mortality rates among cats infected with the FHV-A1 isolate underscore its potential as an ideal candidate for establishing a challenge model and as a potential vaccine strain. However, further investigation comparing the immunogenecity of these FHV isolates with the vaccine strain of an imported triple-inactivated Feline vaccine (Fel-O-Vax^®^ PCT, Zoetis) against prevalent Chinese strains is imperative for developing a more effective vaccine for FHV control in China. This study lays the groundwork for establishing the FHV challenge model and vaccine development.

## 5. Conclusions

Based on the foregoing information, the present study delineates the isolation, characterization, sequencing, phylogenetic analysis, and pathogenicity evaluation of FHV-1 strains isolated from cats in China, underscoring the necessity for ongoing research into upper respiratory tract diseases, particularly FHV-1 in domesticated cats. Our findings revealed significant variability in the severity of respiratory and ocular diseases among cats infected with FHV-1, despite minimal genomic variation observed in these isolates through sequencing and phylogenetic analysis. The FHV-A1 isolate emerges as an ideal candidate for establishing a challenge model and as a potential vaccine strain. Nevertheless, further investigation is warranted to compare the immunogenecity of the FHV isolates with the triple-inactivated Fel-O-Vax^®^ PCT vaccine strain against prevalent Chinese strains, aiming to develop a more effective vaccine for FHV control in China.

## Figures and Tables

**Figure 1 vetsci-11-00285-f001:**
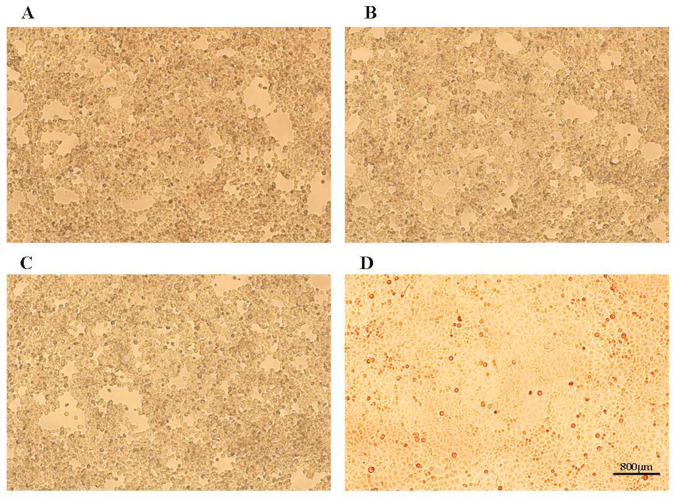
Isolation of the FHV strains in F81 cells. (**A**) Cytopathic effects (CPE) caused by FHV-A1; (**B**) CPE caused by FHV-C8; (**C**) CPE caused by FHV-D3; (**D**) control (uninfected) F81 cells.

**Figure 2 vetsci-11-00285-f002:**
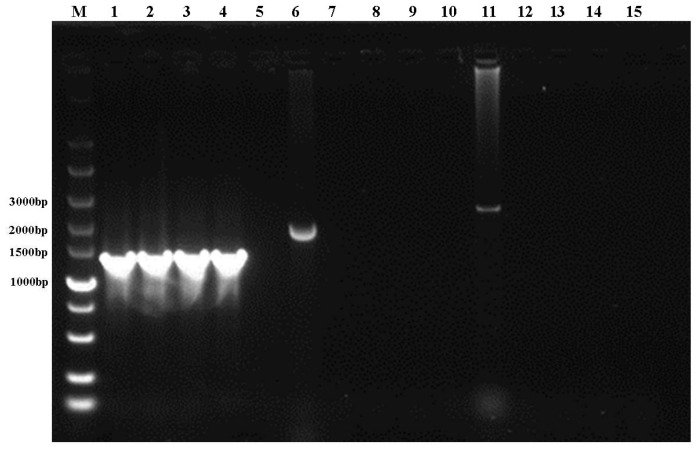
PCR identification of FHV isolates. M: DNA marker; Lanes 1–5 for FHV detection, samples in order: 1: FHV positive control, 2: FHV-A1, 3: FHV-C8, 4: FHV-D3, 5: negative control. Lanes 6–10 for FPV detection, samples in order: 6: FPV positive control, 7: FHV-A1, 8: FHV-C8, 9: FHV-D3, 10: negative control. Lanes 11–15 for FCV detection, samples in order: 11: FCV positive control, 12: FHV-A1, 13: FHV-C8, 14: FHV-D3, 15: negative control.

**Figure 3 vetsci-11-00285-f003:**
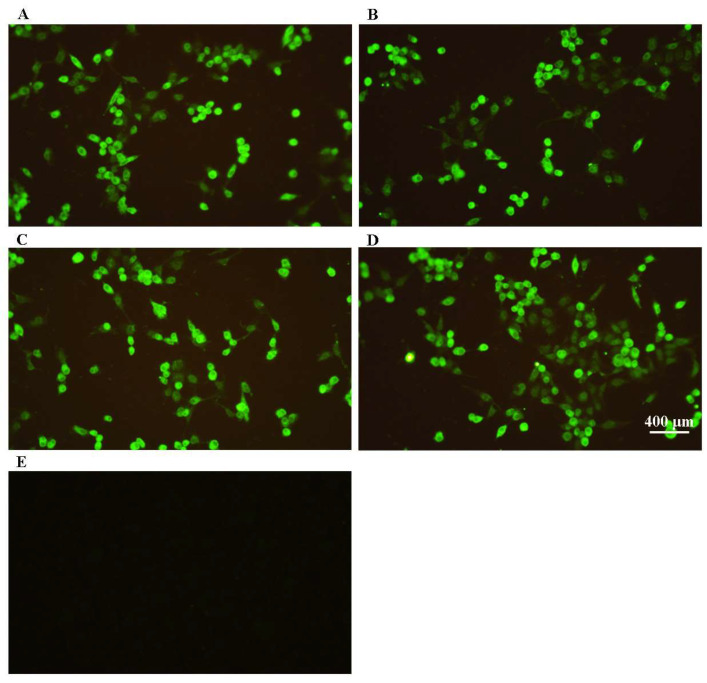
IFA identification of FHV isolates. (**A**) F81 cells infected with FHV-A1; (**B**) F81 cells infected with FHV-C8; (**C**) F81 cells infected with FHV-D3; (**D**) positive control cells; (**E**) normal F81 cells for control.

**Figure 4 vetsci-11-00285-f004:**
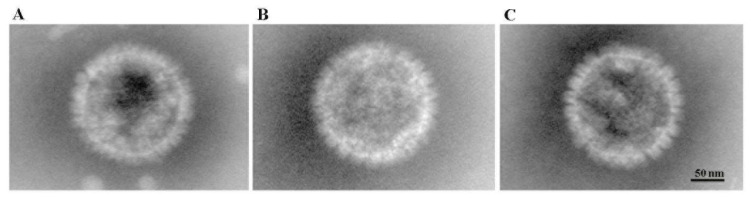
Electron microscopy of FHV isolates. (**A**) Images of FHV-A1 virions; (**B**) images of FHV-C8 virions; (**C**) images of FHV-D3 virions.

**Figure 5 vetsci-11-00285-f005:**
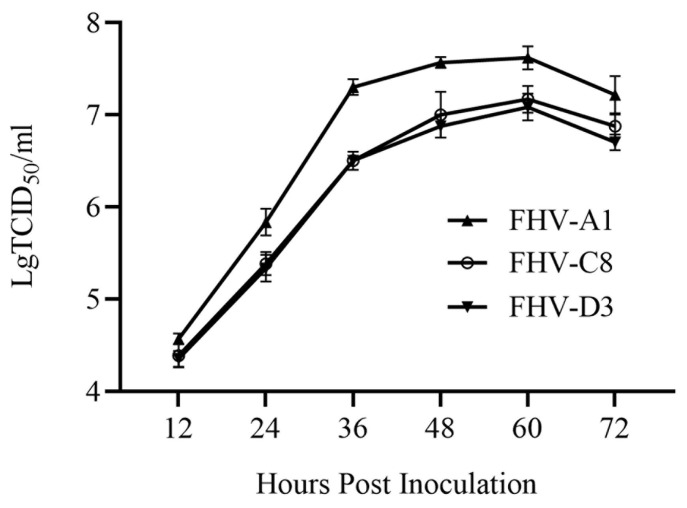
Multiple-step growth curves of the FHV strains (0.01 MOI). Cell cultures were harvested at the indicated time points, and titers were determined on F81 cells. The mean values with standard deviations of three independent experiments are shown.

**Figure 6 vetsci-11-00285-f006:**
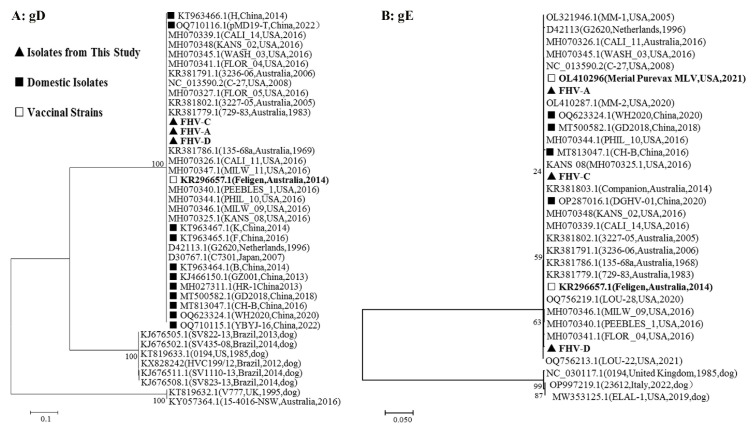
Phylogenetic analysis of nucleotide sequences for FHV *gD* (**A**) and *gE* (**B**) genes.

**Figure 7 vetsci-11-00285-f007:**
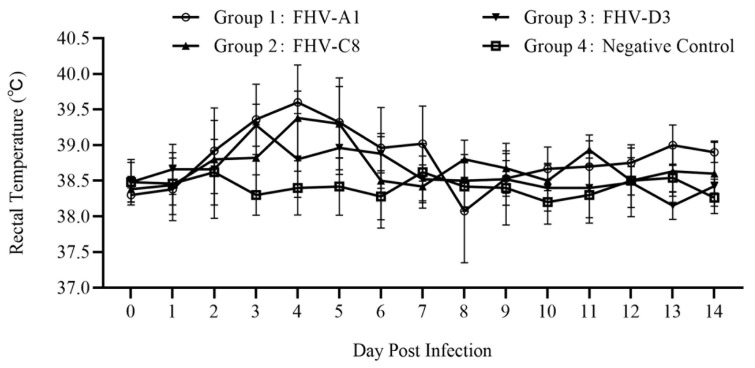
Rectal Temperature Changes in Cats Infected with FHV Isolates.

**Figure 8 vetsci-11-00285-f008:**
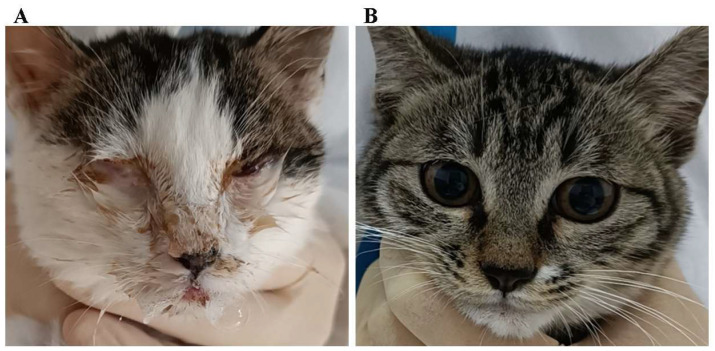
Typical clinical signs after FHV infection. (**A**) FHV-A1 isolate; (**B**) negative control group.

**Figure 9 vetsci-11-00285-f009:**
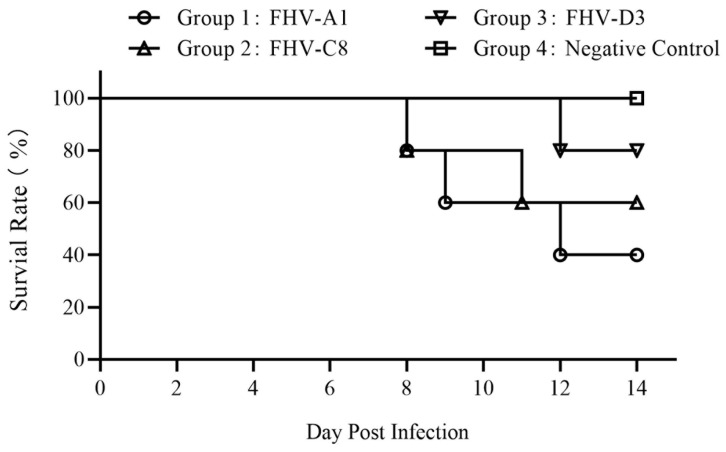
Survival rate of kittens infected with FHV isolates.

**Figure 10 vetsci-11-00285-f010:**
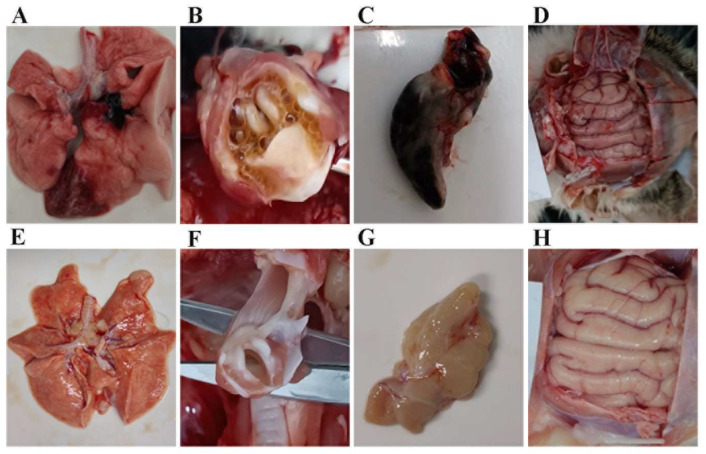
Gross pathological changes after FHV infection (**A**,**E**) lungs; (**B**,**F**) trachea; (**C**,**G**) mesenteric lymph nodes; (**D**,**H**) brain. (**A**–**D**) Kittens that showed severe clinical signs of FHV infection; (**E**–**H**) kittens in the negative control group.

**Figure 11 vetsci-11-00285-f011:**
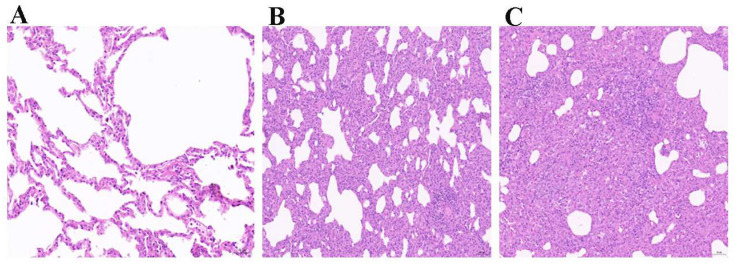
Microscopic pathological changes of the lung after FHV infection. (**A**) Control; (**B**) mild interstitial pneumonia; (**C**) severe interstitial pneumonia. Scale bar = 50 μm.

**Table 1 vetsci-11-00285-t001:** The homology of gene sequences of the FHV Isolates and FHV reference strains.

Gene	Sequence Type	Isolates from This Study	Domestic Isolates	Foreign Isolates
*gB*	Nucleotide	100%	100%	99.9~100%
	Amino acid	100%	100%	99.7~100%
*gD*	Nucleotide	100%	99.9~100%	100%
	Amino acid	100%	99.7%	100%
*gE*	Nucleotide	99.9~100%	99.7~100%	99.8~100%
	Amino acid	100%	99.4~100%	99.6~100%
*TK*	Nucleotide	100%	99.9~100%	100%
	Amino acid	100%	100%	100%

**Table 2 vetsci-11-00285-t002:** Summary of typical clinical signs in kittens infected with FHV isolates.

Group	Inoculum	Fever (≥2 Occasions)	Ocular Discharge, ≥2 Scores Lasting ≥2 Days	Conjunctivitis, ≥2 Scores Lasting ≥2 Days	Nasal Discharge, ≥2 Scores Lasting ≥2 Days	Nasal Congestion, ≥2 Scores Lasting ≥2 Days
1	FHV-A1	5/5	5/5	5/5	5/5	5/5
2	FHV-C8	3/5	3/5	3/5	3/5	3/5
3	FHV-D3	2/5	2/5	2/5	2/5	2/5
4	DMEM	0/5	0/5	0/5	0/5	0/5

**Table 3 vetsci-11-00285-t003:** FHV virus isolation results from nasal and ocular swabs of kittens after FHV isolate infection.

Group	Inoculum	Day Post-Infection (d)														
		0	1	2	3	4	5	6	7	8	9	10	11	12	13	14
1	FHV-A1	0/5	3/5	3/5	5/5	4/5	5/5	5/5	5/5	4/4 ^#^	3/3 ^#^	3/3 ^#^	3/3 ^#^	2/2 ^#^	2/2 ^#^	2/2 ^#^
2	FHV-C8	0/5	1/5	2/5	2/5	5/5	4/5	5/5	5/5	4/4 ^#^	4/4 ^#^	4/4 ^#^	2/3 ^#^	1/3 ^#^	0/3 ^#^	0/3 ^#^
3	FHV-D3	0/5	1/5	2/5	1/5	4/5	5/5	5/5	5/5	4/5	4/5	4/5	2/5	1/4 ^#^	1/4 ^#^	1/4 ^#^
4	DMEM	0/5	0/5	0/5	0/5	0/5	0/5	0/5	0/5	0/5	0/5	0/5	0/5	0/5	0/5	0/5

Note: Numerator indicates the number of kittens that were tested FHV-positive for virus isolation, and the denominator indicates the total number of samples tested. “^#^” indicates the number of other kittens in the group that had died before the sampling.

## Data Availability

All data are included within this article and Appendix A.

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
