# Peer review of "Identification, Genetic Characterization, and Pathogenicity of Three Feline Herpesvirus Type 1 Isolates from Domestic Cats in China"

_vetsci, 2024, doi:10.3390/vetsci11070285_

Round 1

Reviewer 1 Report

Comments and Suggestions for Authors

The work is well designed and its relevance in the field of clinical is evident. The study has evaluated the characteristics and virulence of FHV-1 isolated from symptomatic cats, which is significant for diagnosing, treating, and preventing for Feline herpesvirus in cats. However, the content of the paper requires minor modifications.

1.     The method description is too detailed, it can be appropriately simplified (Section 2.3, 2.5.2 and 2.6).

2.     Page 2, line 70, sentence ‘These cats exhibiting respiratory and ocular symptoms suggestive of herpesvirus infection.’ can be deleted. Line 71, sentence ‘We mainly employed…three strains.’ should be rewrite.

Comments on the Quality of English Language

1.     Page 2, line 48, sentence ‘The clinical outcome of FHV-1 infection can varies…’ should be corrected to ‘The clinical outcome of FHV-1 infection can varie…’.

2.     Page 12, line 366 to 373, this paragraph needs to be reorganized.

3.     Page 12, line 376, sentence ‘Phylogenetic analysis performed …has host selectivity.’ need to be rewrite.

4.     Page 12, line 385, ‘disease was’ should be changed to ‘disease is’.

5.     Page 12, line 389 to 394 should be rewrite.

Reviewer 2 Report

Comments and Suggestions for Authors

Feline herpes is a common feline disease caused primarily by the transmission of feline herpesviruses. Herpesviruses are enveloped, double-stranded DNA viruses that can infect a wide range of animals and remain latently infected for life. Feline herpesvirus (FHV-1) causes upper respiratory disease and ocular lesions in cats, poses a health risk to cats, and may be transmitted to humans. Therefore, it is important to study the transmission route, pathogenicity and control methods of FHV-1. In this study, the authors performed PCR on 25 cat ocular and nasal swab samples, sequencing and phylogenetic analysis of FHV positive samples, and isolation and identification of the virulent strains as well as animal experiments, to get a preliminary grasp of the genomic variation of FHV-1, and to provide a candidate high virulent strain for the study of inactivated vaccine of FHV-1. This study will lay the foundation for pathogenesis, vaccine immunization, diagnostic and molecular biology research of FHV-1. However, there are several issues that need to be addressed. These points are summarized below.

1.In the part of material method , the logic is not clear, it is recommended to reorganize it, and it is best not to appear the three-level title.

2. Line 89, it is best to list the province where the sample was collected.

3. It is suggested that the authors combine 2.5.1 with 2.5.2 and name it as "2.5 PCR and Indirect Immunofluorescence Assay (IFA) Identification".

4.Line 127-128, the reaction system and amplification conditions of PCR should be described briefly.

5. Line 130-131, the amount of virus inoculated needs to be determined, such as "The F81 cells were cultured in 6-well plates until a monolayer was formed, and then the virus was inoculated onto the cells, with two replicates for each sample, at a multiplicity of infection (MOI) of 1", and the "at 100 μl per well" sould be deleted.

6. Line168, "2.5.4Electron microscopy" it better changed as" 2.6 Electron microscopy" .

7. Line154, "2.5.4 Virus Titer and virus growth curve" it better changed as "2.7 Virus Titer and virus growth curve".

8. Line 169, how to extract the viral DNA, please list the method briefly.

9. Line168,"2.6 Sequencing and phylogenetic analysis" it better changed as" 2.8 Sequencing and phylogenetic analysis".

10. Line179,"2.7 Pathogenicity evaluation of FHV isolates in kittens" it better changed as "2.9 Pathogenicity evaluation of FHV isolates in kittens", and combined the two paragraphs 2.7.1 and 2.7.2 into one paragraph.

11. The kittens were inoculated intranasally with 2 mL of of FHV-A1, FHV-C8, and FHV-D3 isolates, respectively. This 2 mL liquid were inoculated in one or more than one time?

12. The picture definition of Figure 4 is not good, it is better to improve the definition.

Comments on the Quality of English Language

The authors did a large amount of experiments, the experimental design is reasonable, and the results of the study are true and credible,the quality of English is better and the writing level is more professional.

Reviewer 3 Report

Comments and Suggestions for Authors

The present work describes the dynamics relating to FHV-1 infection in 35 suspected cats (confirmed with laboratory analysis) at the clinical level, excretion level, and characteristics of the virus (genomic and in vitro replication) responsible for outbreaks in China. The work has a certain methodological rigor, although the number of samples is small and the final aim of the study is unclear. The discussion certainly needs to be improved with references to the most recent bibliography and by comparing the results obtained with those of other studies. I also suggest changing the title of the work, as it is not very representative of what has been done. Below are some of my specific suggestions.

Line 12: The number of samples collected was small (35). Moreover, these samples were collected only in some parts of China (not completely representative). I suggest the authors  change the title, choosing another more representative of the study performed. 

Line 12–14: Please check and rephrase this sentence.

Line 15: Please change "strains" to "them."

Lines 17–20: I do not agree with this statement. Please change the conclusion and state more precisely the contribution of this study to the knowledge of FHV-1 in China.

Line 21: Please check if the sections of the abstract section are permitted.

Line 33-36: As the comment line 17-20

Line 90: Origin of the cells (identifier) Like, for example, the ATCC "code"

Line 98-99: Please repeat this sentence. It appears that the primers were used for PCR, virus neutralization, etc.

Lines 104–110: This part is not clear. I suggest you completely re-evaluate it.

Line 122: Is CPE presented? Maybe there could be better "evident CPE"?

Line 112: I suggest splitting these sentences in two.

Line 126: Please change the subsection title (PCR Identification). Molecular detection?

Line 126: Which kit do you use for DNA exctraction? Calicivirus is an RNA virus. Also, the RNA was exctracted? 

Line 129: Please delete "identification" from the subsection title.

Line 140: I suggest specifying which protein the antibody detects.

Lines 179–200: I suggest making a unique subsection.

Line 209: The authors have to state how they concluded that CPE was caused by other viruses than FHV-1.

Line 220: Figure. In a recent study, Ferrara et al. (2024 found a more evident cytophatic effect using the strain Ba/91 (images are available in the manuscript). I think this could be a good point to improve the discussion section.

Line 260: I suggest another point for the discussion. In another paper concerning the modifications of the PI3K/Akt/mTOR axis during FeHV-1 infection on permissive cells (CRFK), the authors found similar viral titers at different times of infection. 

Discussion: I think this is the main flaw of the manuscript. The discussion section was short and needed improvement. I suggest including other molecular studies that found FHV/FPV/FCV prevalence in cats. For example, recent studies were performed in the Campania region (Italy), the US, Western Canada, etc. I think these studies could be a good starting point to improve the discussion by comparing the prevalences found. 

Comments on the Quality of English Language

The manuscript requires moderate English editing 

Round 2

Reviewer 3 Report

Comments and Suggestions for Authors

The manuscript was significantly improved by the reviewers' comments addressed to the authors as well as by improvements resulting from the English editing. The manuscript is ready to be accepted and published.

Comments on the Quality of English Language

English editing have improved the quality of the manuscript. Enhlish is ok now.